# Dual-gate organic phototransistor with high-gain and linear photoresponse

Philip C.Y. Chow [1,3], Naoji Matsuhisa[1], Peter Zalar[1,2,4], Mari Koizumi[1,2], Tomoyuki Yokota [1,2] & Takao Someya [1,2]

The conversion of light into electrical signal in a photodetector is a crucial process for a wide range of technological applications. Here we report a new device concept of dual-gate phototransistor that combines the operation of photodiodes and phototransistors to simultaneously enable high-gain and linear photoresponse without requiring external circuitry. In an oppositely biased, dual-gate transistor based on a solution-processed organic heterojunction layer, we find that the presence of both *n*- and *p*-type channels enables both photogenerated electrons and holes to efficiently separate and transport in the same semiconducting layer. This operation enables effective control of trap carrier density that leads to linear photoresponse with high photoconductive gain and a significant reduction of electrical noise. As we demonstrate using a large-area, 8 × 8 imaging array of dual-gate phototransistors, this device concept is promising for high-performance and scalable photodetectors with tunable dynamic range.

[1] Department of Electrical and Electronic Engineering, The University of Tokyo, 7-3-1 Hongo Bunkyo-ku, Tokyo 113-8656, Japan. [2] Exploratory Research for Advanced Technology (ERATO), Japan Science and Technology Agency (JST), 2-11-16, Yayoi, Bunkyo-ku, Tokyo 113-0032, Japan. [3] Present address: Department of Chemistry, The Hong Kong University of Science and Technology, Clear Water Bay, Kowloon, Hong Kong. [4] Present address: Holst Centre/ TNO, High Tech Campus 31, 5656 AE Eindhoven, The Netherlands. Correspondence and requests for materials should be addressed to P.C.Y.C. (email: pcyc@ust.hk) or to T.S. (email: someya@ee.t.u-tokyo.ac.jp)

The conversion of light into electrical signal in a photo-detector is a crucial process for a range of applications such as imaging[1-3]. Upon the absorption of photons, electron-hole pairs are generated in a semiconductor and can be separated to form an electrical current. Two classes of photodetectors that have been developed are photodiodes and phototransistors[4]. Photodiodes, which typically operate in reverse bias, create photoresponse by collecting both photogenerated electrons and holes separated by the applied electric fields with high quantum efficiency[5,6]. Due to the lack of intrinsic amplification mechanism, the external circuitry comprising signal amplifiers is typically required for photodetectors based on photodiodes to achieve high signal integrity (for example, integrated complementary metal-oxide-semiconductor CMOS image sensors)[7-10]. The use of external circuitry can be simplified using phototransistors, which provide intrinsic amplification for either photogenerated electrons or holes through photoconductive gain and allows external quantum efficiencies (EQEs) well beyond 100%[11-19]. Photo-conductive gain leads to an increasing EQE with decreasing irradiance (sublinear photoresponse), enabling high-gain detection at low lighting and a broader dynamic range compared to photodiodes[2,12,15]. However, the sublinear photoresponse can be problematic in applications that require high-resolution and quantitative light detection, for which a linear photoresponse with constant EQE provided by photodiodes is preferred[20].

Here we report a new device concept of dual-gate photo-transistor that combines the operation of photodiodes and pho-totransistors to simultaneously achieve high-gain and linear photoresponse without requiring external circuitry. Furthermore, we find that the dual-gate operation significantly reduces the electrical noise in phototransistor, leading to more than three orders of magnitude improvement in photo-detectivity (a measure of the sensitivity of a photodetector). These attributes make dual-gate phototransistor a promising device concept for highly sensitive photodetectors with tunable dynamic range, as we demonstrate using a large-area, 8 × 8 imaging array.

## Results

**Proof-of-concept device.** Figure 1a–d show the device structure and operation of the dual-gate phototransistor in comparison with those of conventional photodiode and phototransistor. In a dual-gate field-effect transistor based on an ambipolar semi-conducting layer, such as an organic bulk-heterojunction (BHJ) blend[21,22], both gates can accumulate and/or deplete both types of charge carriers to form conductive n- and p-type channels in the same layer[23-27]. Since conductive channels are formed within a few nanometres from the gate dielectric-semiconductor interface, two separate channels can be formed by the gates in a sufficiently thick film (>10 nm), such that an n- and a p-type channel can be formed simultaneously when the gates are operating at opposite biases[28], creating an electric field equivalent to the built-in field in diodes for separating electrons and holes.

Our proof-of-concept device comprises a single spin-coated layer of organic BHJ blend (30 nm thick) sandwiched between an indium tin oxide (ITO) bottom gate and a gold top gate, with 70 nm of parylene as the dielectric layer of both gates. Figure 1e shows a schematic illustration of the device structure. Organic BHJ blends consist of interpenetrating networks of n-type and p-type semiconductor, fabricated using solution-processing methods including roll-to-roll printing[29]. These structures are widely used in organic photodiodes to separate photogenerated electron-hole pairs with high quantum efficiencies, and the ease of processing is suitable for low-cost, high-volume and large-area applications[30]. Here we select a model BHJ system that comprises donor polymer: poly[2-methoxy-

5-(3′,7′-dimethyloctyloxy)-1,4-phenylenevinylene] (MDMO-PPV), and fullerene acceptor: phenyl-$C_{61}$-butyric acid methyl ester (PCBM). This blend represents an ideal model system because the photo-carrier generation properties[34] and ambipolar carrier transport on various dielectric layers[22,35,36] have been well characterized in previous studies, and fast photoresponse in a single-gate phototransistor configuration has been demon-strated[13]. Figure 1f shows a top-view optical microscopy image of the device and the electrodes in use. Light is able to reach the photoactive semiconducting layer through the semi-transparent ITO/parylene bottom gate. The source and drain electrodes are patterned using lift-off photolithography, with a total channel width of 100 mm and a length of 5 μm. The effective photoactive area is formed in the overlapping area between the channel and the gate electrodes (red dashed box).

**Quasi-static electrical output and photoresponse.** We first characterize the quasi-static electrical output of the dual-gate organic phototransistor (DGOPT). Figure 2a, b show the transfer and output curves measured in the dark and at various bias conditions. Without top gate bias ($V_{TG} = 0$ V), the device exhibits unipolar, n-type transport as bottom gate ($V_{BG}$) scans between 0 and 20 V[13,22]. At increasing $V_{TG}$ with opposite sign (from 0 to −20 V), we observe two changes to the transfer curves. First, the threshold voltage for the n channel shifts towards more positive $V_{BG}$, leading to a decrease in drain current at any given $V_{BG}$ (up to an order of magnitude at $V_{BG} = 20$ V). Previous reports show that the control of threshold voltage is resulted by the electrostatic interaction between the two accumulation layers[23-28]. As dis-cussed by Roelofs et al. using a numerical drift-diffusion model[28], a negative $V_{TG}$ depletes the n channel accumulated by the bottom gate in a sufficiently thin (<50 nm) semiconducting layer, thus leading to an increase in threshold voltage. Second, we observe increasing drain current at low $V_{BG}$ when $V_{TG}$ is more negative than about −10 V, and this increasing current is much greater than leakage through the gate electrodes (see Supplementary Fig. 1–4). We therefore assign this to the opening of a p channel due to the accumulation of holes at the top dielectric interface when a sufficiently negative $V_{TG}$ is applied, resulting in ambipolar carrier transport through two separated, but electrostatically interacting, conductive channels in the same semiconducting layer.

Upon light illumination, a photocurrent ($I_{photo} = I_{light} − I_{dark}$, where $I_{light}$ and $I_{dark}$ are drain currents in light and dark, respectively) is created at all $V_{BG}$. Figure 2c shows the transfer curves measured at a range of light intensities at $V_{TG} = 0$ V (top) and $V_{TG} = −20$ V (bottom). The increase in drain current by light can be interpreted as an increase in either electron (for n-type) or hole (for p-type) accumulation following the separation of photogenerated electron-hole pairs, thus resulting in a reduced threshold voltage[11,14]. The photoconductive gain[2] of a photo-transistor, G, scales with the ratio of carrier trapping lifetime, $\tau_{trap}$, and the transit time of mobile carriers (with opposite sign) across the accumulation channel, $\tau_{transit}$,

$$G = \tau_{trap}^{e,h} / \tau_{transit}^{h,e} \qquad (1)$$

At $V_{TG} = 0$ V, there is no p channel and thus photogenerated holes remain trapped, either in the bulk of the semiconducting blend or at the dielectric-semiconductor interfaces[4], leading to photoconductive gain. The photocurrent is largest when the channel is accumulated with electrons ($V_{BG} = 20$ V), at which the photogenerated electrons are strongly amplified. Figure 2d shows the intensity dependence of the photocurrent and the corre-sponding photo-responsivity (defined as $R = I_{photo}/P_{in}A$, where $P_{in}$ is the power of the incident light per unit area and A is the

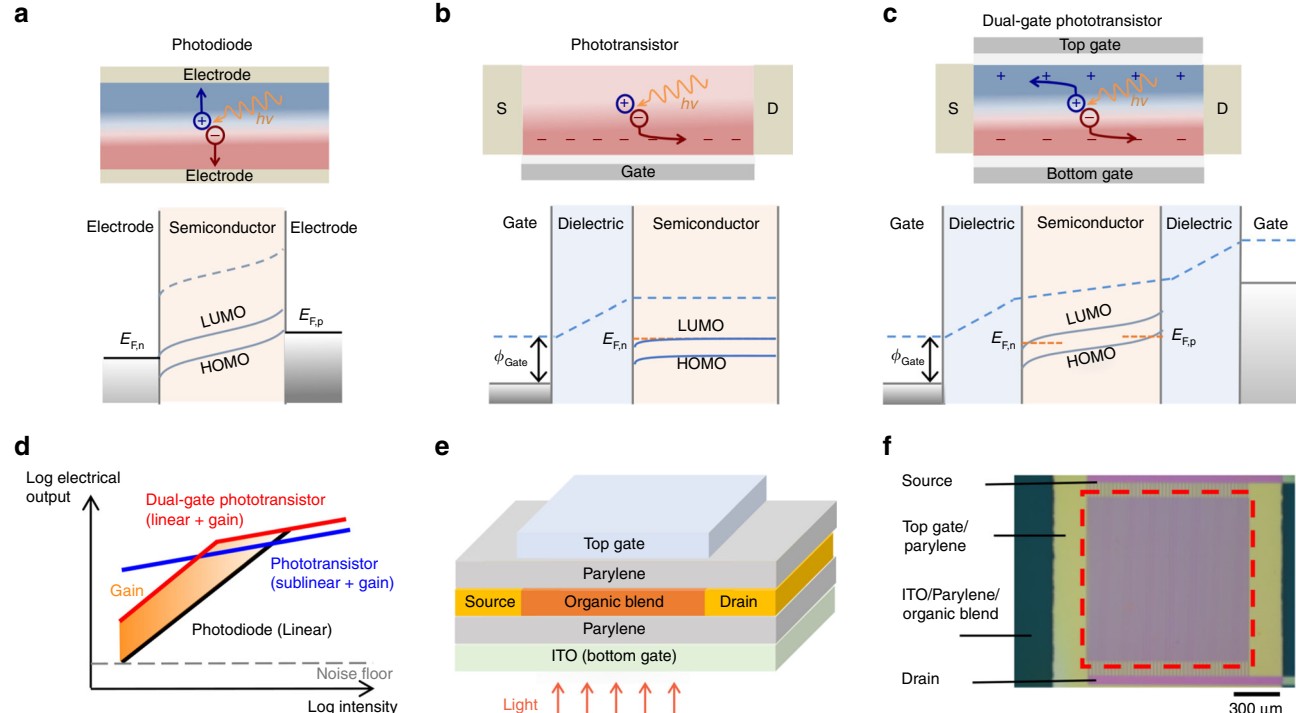

**Fig. 1** Existing photodetectors and introducing the new device concept of dual-gate phototransistor. **a–c** Schematic illustration and explanatory band diagrams of photodiode, phototransistor, and dual-gate organic phototransistor (with opposite gate biases), respectively. Electron-holes pairs are generated in the semiconducting layer upon light absorption, and are separated to form a photocurrent. Photodiodes can enhance charge separation and collect both carrier types by applying an electric field between the electrodes. Phototransistors only collect one type of carriers by accumulating either electrons or holes at the dielectric-semiconductor interface to form a conductive channel, while the opposite charge remains trapped to allow photoconductive gain. Dual-gate phototransistor consists of two accumulation channels, which can simultaneously conduct opposite charges when the gates are oppositely biased in a sufficiently thick (>10 nm) semiconducting film. The electrostatic interaction between the two accumulation layers introduces an electric field which separates electrons and holes, similar to photodiodes. S/D denotes source and drain electrodes in field-effect transistor architecture. **d** Dynamic range of a photodetector based on photodiodes (linear photoresponse), phototransistor (sublinear photoresponse with photoconductive gain), and dual-gate phototransistor (linear photoresponse with photoconductive gain until channel saturation). **e** Device structure of dual-gate phototransistor based on bulk-heterojunction blend of MDMO-PPV and PCBM at 1:15 weight ratio. A self-assembled monolayer of 1-dodecanethiol was formed on the gold source/drain electrodes to improve charge injection. Light is able to reach the photoactive BHJ layer through the semi-transparent ITO/parylene bottom gate. **f** Top-view optical microscopy image of the device and the electrodes in use. The source/drain electrodes were patterned using lift-off photolithography into a comb-shaped interdigitating pattern, with a total channel width and length of 100 mm and 5 µm, respectively. The effective photoactive area is formed in the overlapping area between the channel and the gate electrodes (red dashed box)

device area). The photocurrent at $V_{TG} = 0$ V scales sublinearly (slope ~ 0.3 in logarithmic scale) with intensity. This dependence is commonly found in phototransistors, in which the high-gain photoresponse is due to the occupancy of energetically distributed trap states[12,16]. At low intensities, the long-lived, low-energy trap states are preferentially populated to allow high photoconductive gain. But at higher intensities, these trap states are saturated, and therefore shorter-lived, energetically shallow trap states are populated instead, resulting in a decreasing gain with increasing intensity that leads to a sublinear photoresponse.

At $V_{TG} = -20$ V and $V_{BG} = 20$ V, we find that the photo-current exhibits different linearity in two intensity regimes. We note that at these operation biases the drain current is mainly contributed by electrons in the bottom $n$ channel and only partly contributed by holes in the top $p$ channel (as reflected by the transfer curve in Fig. 2a, c). At high intensities (> 1 mW cm$^{-2}$), the linearity of the photocurrent with respect to intensity is not changed by $V_{TG}$. In this intensity regime, the photoresponse scales sublinearly with intensity regardless of the top gate bias (see Supplementary Fig. 5). This indicates that carrier trapping-induced photoconductive gain is the dominant effect at high intensities at both $V_{TG}$ conditions. We also note that the photocurrent increases by about a factor of two when the gates

are operating at opposite biases in this intensity regime. This is likely due to improved separation of photogenerated electron-hole pairs by the vertical electric field introduced by the oppositely biased gates. At lower intensities (<1 mW cm$^{-2}$), however, we observe a change in slope and the photocurrent scales linearly with intensity (slope = 0.97 ± 0.05). Such linear photoresponse is reflected in the near constant responsivity of about 1 AW$^{-1}$, which corresponds to an external quantum efficiency (EQE) of ~220% at 543 nm[4]. According to Equation 1, a constant gain is achieved by maintaining the ratio between carrier trapping and transit times. Previous reports have shown that carrier mobility (and thus, carrier transit time) of dual-gate transistors is minimally affected by the top gate bias[24]. Therefore, a nearly constant carrier trapping time is required to maintain a constant gain when we operate with a negative top gate bias.

We consider that the dual-channel operation at oppositely biased top/bottom gates is the key to achieving such a linear photoresponse with constant amplification. As illustrated in Fig. 1c, the oppositely biased gate electrodes form both $n$ and $p$ channels in the same semiconducting layer. At $V_{TG} = 0$ V, photogenerated electrons drift across the $n$ channel, while the holes remain trapped to provide photoconductive gain. However, at negative $V_{TG}$, the creation of the $p$ channel provides a pathway

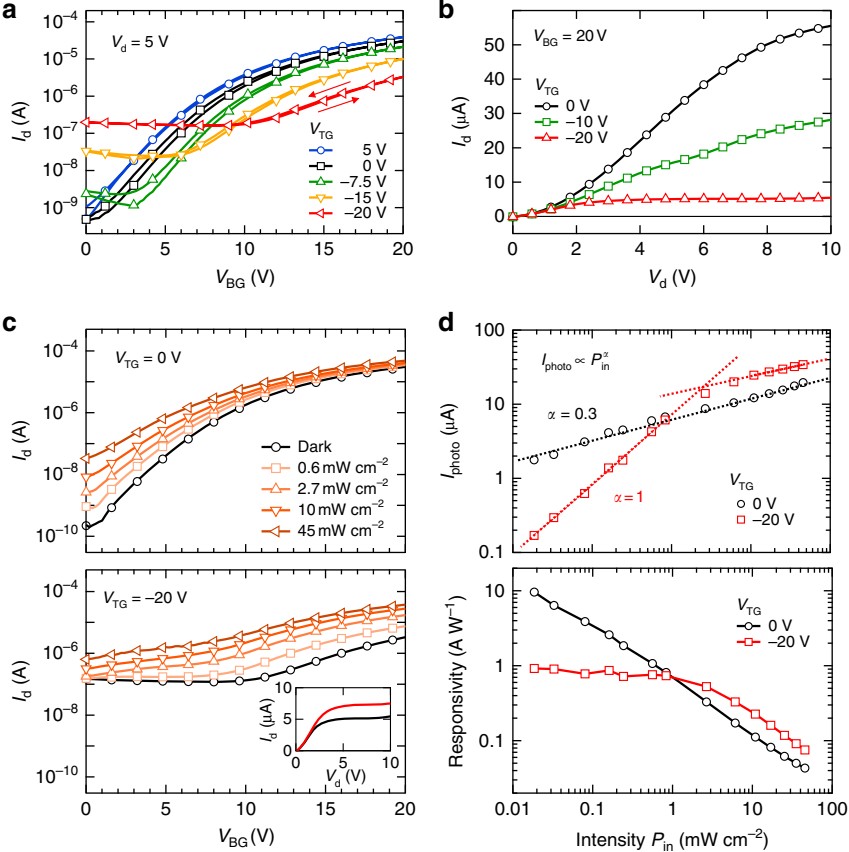

**Fig. 2** Quasi-static optoelectrical characterisation of dual-gate organic phototransistor. **a** Transfer curves measured in dark at various top gate biases ($V_{TG}$), with fixed source-drain bias ($V_d = 5V$). The arrows indicate the scanning direction. The increasing drain current at low $V_{BG}$ indicates the formation of a $p$ channel due to the accumulation of holes at the top dielectric interface when a sufficiently negative $V_{TG}$ is applied, and the threshold voltage shift of the $n$ channel indicates that the two channels are electrostatically interacting. At $V_{BG} = 20$ V and $V_{TG} = -20$ V, the drain current is mainly contributed by electrons in the bottom $n$ channel and only partly contributed by holes in the top $p$ channel. **b** Output curves measured in dark at various top gate biases ($V_{TG}$), with fixed bottom gate bias ($V_{BG} = 20$ V). **c** Transfer curves measured at various light intensities with $V_{TG} = 0$ V (top) and $V_{TG} = -20$ V (bottom), at $V_d = 5$ V. A positive photocurrent ($I_{photo}$) was generated at all gate bias conditions due to the shift in threshold voltage induced by light. Inset shows the output curve in dark (black) and in light (red) at 0.5 mW cm$^{-2}$ irradiance at $V_{TG} = -20$ V and $V_{BG} = 20$ V. **d** Logarithmic plots showing the dependence of photocurrent (top) and responsivity (bottom) on light intensity ($P_{in}$) at both top gate bias conditions (fixed $V_{BG} = 20$ V). At $V_{TG} = 0$ V, photocurrent scaled sublinearly with intensity throughout the entire range displayed ($I_{photo} \propto P_{in}^{-\alpha}$, with slope $\alpha = 0.28 \pm 0.03$). At $V_{TG} = -20$ V, photocurrent switched from sublinear dependence at high intensities (>1 mW cm$^{-2}$, slope $\alpha = 0.32 \pm 0.04$) to linear dependence at low intensities (<1 mW cm$^{-2}$, slope $\alpha = 0.97 \pm 0.05$). The dotted lines show the power-law fits. The linear scaling of photocurrent with light is reflected by the constant high-gain responsivity of about 1 AW$^{-1}$, which corresponds to an external quantum efficiency (EQE) of ~220% at 543 nm. Similar results were obtained in a dual-gate phototransistor made of DPP-DTT:PCBM blend, but with much-improved performance (~40 AW$^{-1}$ or ~9000% EQE) due to the enhanced carrier mobility in this blend (see Supplementary Fig. 9)

for photogenerated holes to be extracted. As a result, in the linear responsivity regime, additional holes generated in the device do not contribute to the trap carrier density and trapping time becomes independent of light intensity, leading to a constant photoconductive gain. The gain is, therefore, primarily driven by low-energy, deep trap states that are not collected by the $p$ channel. The existence of these states is supported by the sublinear responsivity exhibited at very low light intensities (<10 μW cm$^{-2}$, see Supplementary Fig. 6), at which all the photoexcited holes populate these low-energy trap states[16]. In this excitation range, the reduced photoconductive gain found in the dual-channel regime indicates that the carrier trapping time is shorter as a result of field assisted processes from the negative top gate bias, but still sufficiently long to provide gain. At high intensities (>1 mW cm$^{-2}$), the sublinear response is due to saturation of the bottom channel, resulting in a linear dynamic range of nearly 2 orders. We find that this range can be significantly increased to nearly 4 orders by further depleting the

bottom channel with the top gate bias (see Supplementary Fig. 7), although in the expense of gain. Further description of the working mechanism is found in the SI (see Supplementary Fig. 8).

We further study the working mechanism of DGOPT by replacing the organic layer with a BHJ blend of poly($N$-alkyl diketopyrrolo-pyrrole dithienylthieno [3,2-$b$]thiophene) (DPP-DTT) polymer and PCBM[12]. In contrast to MDMO-PPV:PCBM, this system has high hole mobility ($p$-type) and therefore photoconductive gain is enabled by hole transport and electron trapping. By applying opposite biases on the top and bottom gates (negative $V_{BG}$ and positive $V_{TG}$), we observe linearized photo-response at a similar intensity range as observed for the MDMO-PPV device, but with significantly greater responsivity values (~ 40 AW$^{-1}$ or ~ 9000% EQE) due to the enhanced carrier mobility in the DPP-DTT blend (see Supplementary Fig. 9). This result confirms that either electrons or holes can be transported across the device to provide amplified and linearized photoresponse, provided that the opposite charge can be extracted through the

top channel to maintain a constant trapping density with respect to light intensity (and avoid channel saturation). Moreover, this result confirms the universality of this new device concept, and shows that DGOPT with high gain and linearized photoresponse can be achieved by using high mobility materials.

**Dynamic photodetection performance.** We turn to study the photodetection performance of DGOPT. Figure 3a shows the transient response of our device illuminated with a pulsed light source (543 nm, 10 mW cm$^{-2}$) modulated at 10 Hz using an optical chopper. As discussed, at $V_{TG} = -20$ V the dark current drops by an order of magnitude and the photocurrent increased by a factor of 2 at this intensity regime. Figure 3b shows the normalized photocurrent. At high intensity (7.5 mW cm$^{-2}$), the device operating at a top gate bias of 0 V ($-20$ V) exhibits a 10-90% rise time of 8 ms (15 ms) and a 90–10% decay time of 34 ms (27 ms). The rise time is associated with the time taken for charge trapping/detrapping processes to reach steady state upon illumination, and the decay time corresponds to the time taken for charges to detrap in the dark. At opposite gate biases, the slower rise time is likely due to the increase in electron-hole separation by the vertical field and dual-channel operation, and the faster decay time can be explained by the sweep out of trapped holes by the top $p$ channel which is not allowed at $V_{TG} = 0$ V. The same trend was observed at lower intensity (0.5 mW cm$^{-2}$), but with reduced response times at both biases such that the photoresponse is RC time limited at 10 Hz (Supplementary Fig. 10). This is consistent with the presence of energetically distributed

trap states, with low-energy, longer-lived states preferentially filled at lower intensities[12,16].

A key figure of merit for photodetectors is specific detectivity $D^*$, defined as[2,5]

$$D^* = R\sqrt{A}/I_N \tag{2}$$

where $R$, $A$, and $I_N$ are responsivity, device area, and noise current spectral density (noise current, in unit of A Hz$^{-1/2}$), respectively. Figure 3c shows the responsivity, noise current and the calculated specific detectivity as a function of frequency at various top gate biases. Responsivity decreases with increasing frequency due to the build-up of traps, and we find little difference between device responsivity with/without applying top gate bias at 1 mW cm$^{-2}$ in the displayed frequency range. However, we detect a significant reduction in noise current with increasingly negative bias (up to 3 orders). All the noise measurements were performed in the dark, as described in the Methods section. Capture and release of charge carriers by trap sites, causing carrier mobility and/or number fluctuations, is one of the major sources of noise in semiconductors, commonly referred to as $1/f$ or flicker noise[32]. We find that the noise power spectra of DGOPT show clear $1/f$ dependence with slope ~1 (Supplementary Fig. 11), and its reduction by negative top gate bias is attributed to the decrease in drain current (Fig. 2) and carrier fluctuation. This drastic noise reduction translates to a major improvement in the photodetection performance, with detectivity beyond $10^{10}$ Jones (cm Hz$^{1/2}$W$^{-1}$) at a top gate bias of $-20$ V and 1 mW cm$^{-2}$ irradiance (compared to ~$10^7$ Jones at $V_{TG} = 0$ V at the same intensity).

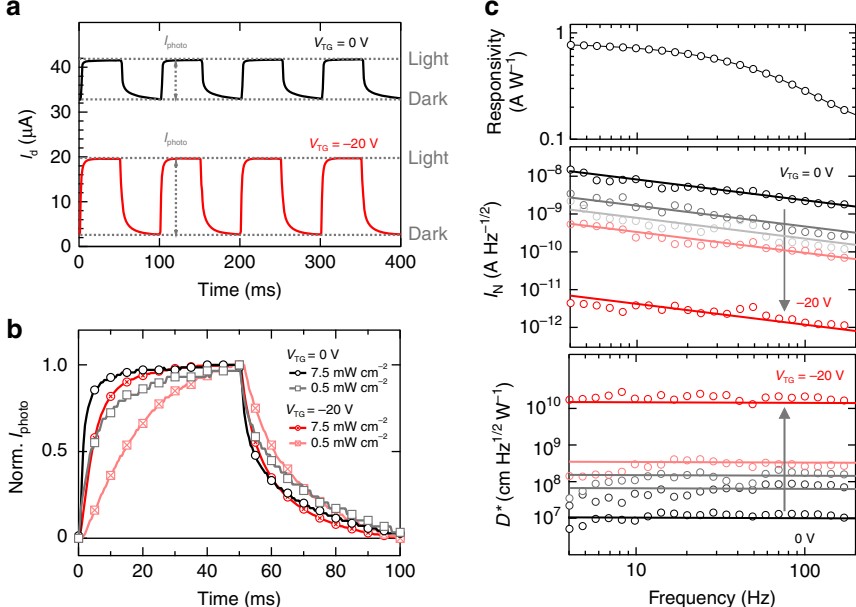

**Fig. 3** Photodetection performance of dual-gate organic phototransistor. **a** Transient response of drain current with and without top gate bias ($V_{TG} = 0$, $-20$ V), operating at fixed source-drain bias of 5 V and $V_{BG}$ of 20 V. A monochromatic light source was employed for excitation (543 nm, 10 mW cm$^{-2}$), modulated at 10 Hz using a mechanical chopper. At such high intensity, the twofold increase in photocurrent is likely due to the improved separation of photogenerated electron-hole pairs by the vertical electric field introduced by the oppositely biased gates. **b** Normalized photocurrent showing the rise and decay kinetics at high (7.5 mW cm$^{-2}$) and low (0.5 mW cm$^{-2}$) intensities. At both biases, slower response time was observed at low intensity, which is consistent with the filling of energetically distributed trap states. With negative $V_{TG}$, slower rise time is likely due to the increase in electron-hole separation by the vertical electric field which extends time taken for the charge trapping/detrapping processes to reach steady state. In dark, the faster decay time can be explained by the sweep out of trapped holes by the top $p$ channel which is not allowed at $V_{TG} = 0$ V. **c** Photo-responsivity, noise current spectral density $I_N$ and specific detectivity $D^*$ as a function of frequency at various $V_{TG}$. Responsivity, at 1 mW cm$^{-2}$, was reduced with increasing frequency since it is limited by the response times, and little difference was observed with/without negative $V_{TG}$. $I_N$ also decreased with increasing frequency, showing $1/f$ characteristics. A significant reduction in $I_N$ was observed at increasingly negative $V_{TG}$ (3 orders of magnitude from 0 to $-20$ V). Such improvement in noise is also reflected in the resulted specific detectivity, which is nearly constant in the displayed frequency range since the reduction in responsivity is compensated by the improved noise. The solid lines serve as guides for the eye

Although the detectivity value of our proof-of-concept device is relatively low (~$10^{12}$ Jones for silicon diodes)[5], we note that $10^{10}$ Jones detectivity at a relatively high irradiance of 1 mW cm$^{-2}$ is amongst the best achieved for organic phototransistors (as summarized in Supplementary Table 1). Additionally, at low irradiance (<100 nW cm$^{-2}$), the oppositely-gated DGOPT device exhibits sublinear photoresponse and thus high detectivity up to $1.5 \times 10^{12}$ Jones is achieved owing to the much-reduced noise. Therefore, the dual-gate operation of organic phototransistor provides a promising pathway towards achieving high-gain and linear photoresponse at higher irradiance without losing the high detectivity at low irradiance typically found in conventional phototransistors. We also note that the observed noise current is considerably higher than the shot noise from the drain current (Supplementary Fig. 12), and therefore an overestimated specific detectivity value is obtained if we assume that the shot noise equals the total noise current.

**Imaging sensor**. We demonstrate the potential use of DGOPT for highly sensitive and large-area imaging applications. Figure 4 shows a two-dimensional 8 × 8 array of DGOPT devices used for imaging light (1.2 mW cm$^{-2}$) passing through a semi-transparent (40% at 540 nm) polyimide shadow mask with a T-shaped pattern (1.5 cm wide). Without applying $V_{TG}$, the pattern cannot be resolved due to the high level of noise and sublinear photoresponse, which generates large background currents (~3 µA)

passing through the semi-transparent area of the shadow mask. On the other hand, we achieve significant enhancement of image resolution by applying negative $V_{TG}$ as a result of reduced noise and linear photoresponse. We apply a slightly lower negative $V_{TG}$ (−17.5 V instead of -20 V) to limit cross-talk between pixels that share common gate electrodes (Supplementary Fig. 13). At this operation condition the linear photoresponse creates sufficient contrast between the image and the background (~0.3 µA), enabling the image to be resolved.

## Discussion

Our results show that DGOPT operates as a lateral-structured photodiode, where the photoresponse is created upon the collection of photogenerated carriers in both $n$ and $p$ channels, but with the addition of an intrinsic amplification mechanism through photoconductive gain. Furthermore, the electrostatic interaction between the two accumulation channels with opposite charge polarity creates an electric field that aids charge separation, which is analogous to the built-in field in photodiodes[28]. We note that a similar high-gain and linear photoresponse is demonstrated in a hybrid photodetector of an integrated colloidal quantum dot photodiode directly atop a graphene transistor[31], which rely on electron/hole dissociation in the photodiode and hole amplification in the adjacent transistor following charge transfer. Additionally, similar achievement is reported in a nanocomposite photodetector composed of ZnO nanoparticles

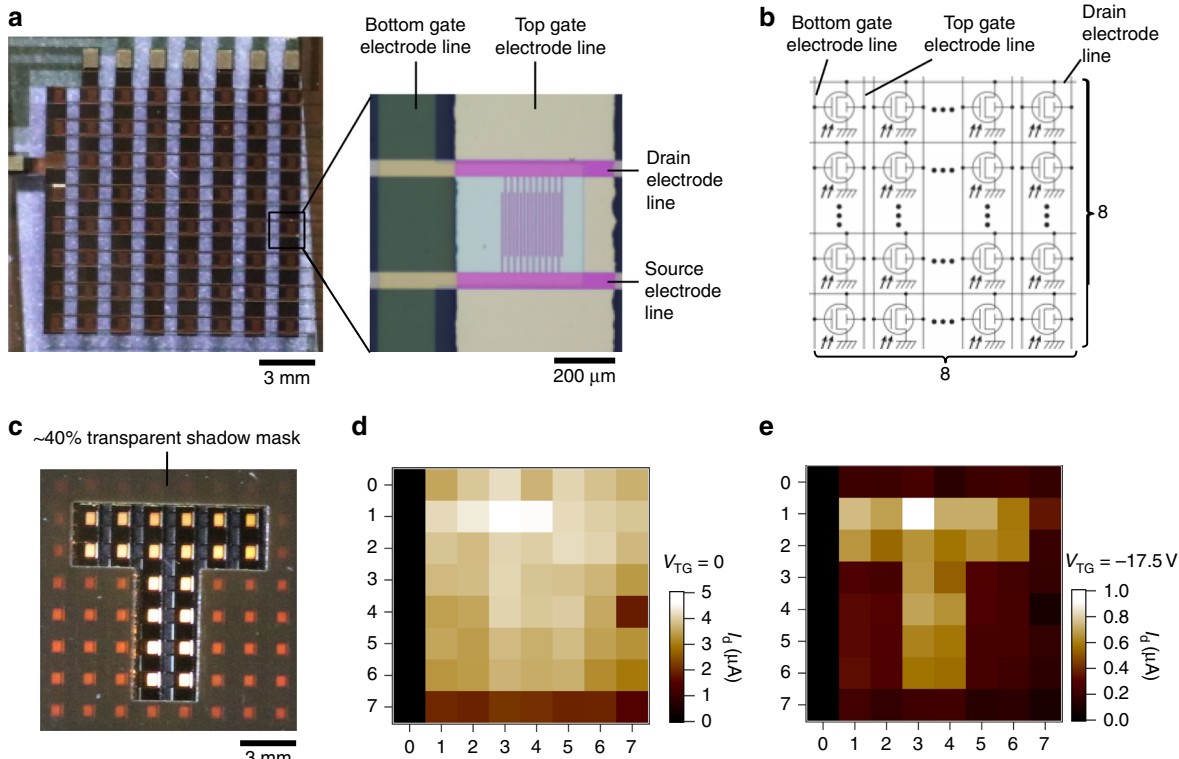

**Fig. 4** Two-dimensional 8 × 8 array of dual-gate organic phototransistors for imaging. **a** Optical micrograph showing the arrangement of pixels. Each individual pixel has the same structure as shown in Fig. 1e, with channel length and width of 5 µm and 5 mm, respectively. Each column containing 8 pixels shares a common top gate electrode of gold and a common bottom gate electrode of indium tin oxide. **b** Circuit schematic of the image sensor. **c** The array was illuminated with white light (1.2 mW cm$^{-2}$ at 543 nm) through a semi-transparent (~ 40% transmittance at 543 nm), polyimide shadow mask with T-shaped pattern. **d** At $V_{TG} = 0$ V, the pattern could not be resolved due to a large background signal (~ 3 µA) caused by light passing through the semi-transparent shadow mask (0.5 mW cm$^{-2}$). The large background signal is the result of sublinear photoresponse and high level of noise observed in this operation mode. We note that the lack of signal output from column 0 is due to damaged pixels. However, when operating with a negative top gate bias (right), the pattern is clearly imaged (**e**). The negative top gate bias is limited to −17.5 V (instead of −20 V as demonstrated for individual pixels) in order to reduce cross-talk between pixels from the same column. The improvement in image resolution is due to the reduced noise and linear photoresponse at negative top gate bias, limiting the background signal to only about 0.3 µA in this demonstration

blended with semiconducting polymer with interfacial trap-controlled charge injection[16]. The performances of these devices are remarkably high, yet their application could be limited by material selectivity and the need for processing two different components.

We expect that the detectivity of our proof-of-concept DGOPT device can be further improved by optimising device parameters and dimensions. In particular, as we demonstrated with the DPP-DTT device, the device responsivity can be much improved by using materials with either fast hole or electron transport (but still able to create dual transport channels to realize linear photoresponse), and sufficient trapping density for the opposite carriers to enable high gain (as described in Equation 1). Operating mainly in $n$ mode, the MDMO-PPV:PCBM blend used herein exhibits modest responsivity values as a result of its relatively low electron mobility (in range of $10^{-3}$ cm$^2$ V$^{-1}$ s$^{-1}$) that lead to long transit times[13]. Currently, organic semiconductors with high carrier mobilities of over 1 cm$^2$ V$^{-1}$ s$^{-1}$ are available[33], and these may be suitable materials for developing high-performance DGOPT devices. However, we note that devices operating at high drain currents are also likely to have higher noise current which limits detectivity (Equation 2). Therefore, an overall improvement in detectivity requires higher responsivity while maintaining low level of noise.

Our imaging sensor demonstration highlights the importance of linearised photoresponse for high-resolution imaging applications. Although the sublinear photoresponse of conventional phototransistors has lower resolution, it is, however, ideal for detecting very weak light signals (<0.1 µW cm$^{-2}$) due to the high-gain. This has been recently demonstrated by Pierre et al. using a single-carrier, charge-integrated phototransistor to image an opaque shadow mask at low light intensites[15]. Our results show that the linearity of DGOPT can be selectively tuned by the top gate bias (Fig. 2d), and therefore it exhibits a broad dynamic range with tunable resolution to suit different applications. For example, a linear response at negative $V_{TG}$ is selected for high-resolution imaging, and a sublinear response at zero $V_{TG}$ is selected for high dynamic range imaging. In summary, the capability of DGOPT in achieving high-gain and tunable (linear/sublinear) photoresponse without requiring external circuitry makes it a promising technology for high-performance, large-area and scalable optoelectronic sensors.

## Methods

**Device fabrication**. Precleaned ITO on glass (Sumitomo Chemical Co., Ltd.) was used as the device substrate. A 70 nm-thick layer of parylene (dix-SR, Daisankasei Co., Ltd.) was deposited on the ITO by chemical vapour deposition to form the bottom gate. Gold source and drain electrodes were thermally evaporated and patterned using lift-off photolithography, with a total channel width of 100 mm and a length of 5 µm. The gold contacts were submerged in 1-dodecanethiol (Sigma Aldrich, 3 mM diluted in ethanol) for 18 hours to form a self-assembled monolayer to improve injection. A 30 nm-thick blend of poly[2-methoxy-5-(3′,7′-dimethyloctyloxy)-1,4-phenylenevinylene] (MDMO-PPV, Sigma Aldrich) and PCBM (Nano-C) was spin-coated on top of parylene/ITO inside a N$_2$-filled glovebox (weight ratio of 1:15, 10 mg ml$^{-1}$). Another 70 nm-thick layer of parylene and 100 nm-thick layer of gold were deposited to form the top gate. For the alternative blend, precleaned Si/SiO$_2$ (200 nm) wafer was used as the bottom gate. A 25 nm-thick blend of poly($N$-alkyl diketopyrrolo-pyrrole dithienylthieno [3,2-$b$]thiophene) (DPP-DTT, Hydrus Chemical Inc.) and PCBM inside a N$_2$-filled glovebox (weight ratio of 1:1, 10 mg ml$^{-1}$). Gold contacts were thermally evaporated through a shadow mask with a total channel width of 700 µm and a length of 30 µm. A 300 nm-thick layer of parylene and 30 nm-thick layer of gold were deposited on top to form the top gate.

**Optoelectrical characterization**. Electrical characterisations were carried out with a Semiconductor Parameter Analyser (Agilent B1500) under ambient laboratory conditions. A halogen lamp was used as the light source for quasi-static measurements. A set of neutral density filters was used for changing the light intensity. For transient measurements, light from a green HeNe laser (Newport Corporation, 543 nm) was modulated with an optical chopper at 10 Hz. Light intensity was characterised using a bench top optical power metre (Newport Corporation). Noise

spectral density measurement was carried out using a Semiconductor Parameter Analyser (Agilent 4155 C) and Signal Source Analyzer (Agilent E5052B).

**Imaging sensor**. The imaging sensor contains $8 \times 8$ dual-gate organic phototransistors sharing bottom and top gate electrode line in each column, and source and drain electrode line in each row. Each individual pixel consists of a MDMO-PPV:PCBM-based dual-gate phototransistor with device structure as described above, with channel width of 5 mm and a length of 5 µm. Gold source and drain electrodes were thermally evaporated and patterned using lift-off photolithography, and parylene dielectric layers (70 nm) were deposited by chemical vapour deposition. Precleaned ITO on glass was used as the device substrate, acting as the bottom gate, and thermally evaporated gold on the top dielectric layer form the top gate. Images were obtained by measuring each pixel using a Semiconductor Parameter Analyser (Agilent B1500). For the target pixel, a $V_{BG}$ of 20 V, $V_{TG}$ of 0 or $-17.5$ V and $V_d$ of 5 V were applied. Cross-talks between pixels were eliminated by grounding all the other unselected electrode lines.

## Data availability
The authors declare that all data supporting the findings of this study are available from the corresponding author on request.

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

## Acknowledgements

This work was financially supported by the JST ACCEL Grant Number JPMJMI17F1, Japan. P.C.Y.C. is grateful to the Japan Society for the Promotion of Science (JSPS) for a Postdoctoral Fellowship for Overseas Researchers. N.M. is supported by Advanced Leading Graduate Course for Photon Science (ALPS) and the JSPS research fellowship for young scientists.We thank Prof. Henning Sirringhaus (University of Cambridge, UK) and Dr. Sunghoon Lee (University of Tokyo, Japan) for insightful discussions.

## Author contributions

P.C.Y.C., N.M., P.Z., M.K. and T.Y. fabricated and tested the devices; P.C.Y.C, N.M. and T.Y. analysed and interpreted the data; P.C.Y.C., N.M. and T.S. wrote the manuscript. T.S. supervised and directed this project.

## Additional information

**Competing interests:** The authors declare no competing interests

