## [Peer Review file · Nature Communications]

Reviewers' comments:

Reviewer #2 (Remarks to the Author):

The authors answered to my concerns providing very reasonable explanations, and their amendments help to clarify some aspects of their interesting work. Apart from the detail on the bias of the transistor characterized in Fig. S4 (by changing the sign to the drain voltage they are actually inverting the current flow, largely complicating the interpretation of the data), I do not have further technical points.

The working mechanism, relying on a substantial trapping of one of the two carriers, leaves me some doubt whether the device should be really indicated as ambipolar, or instead as a substantially unipolar device provided with a trapping phase for the complementary carrier.

The answers confirm that a proof-of-concept dual-gate phototransistor device is proposed for linearization of the photoresponse with intensity, leaving large room for improvement. The PPV:PCBM choice is based on its use by Anthopoulos long ago. It is reasonable, but the achievement of improved performances (such as an extended linearization) is only postulated and is left to further work. Given the complex combination of charge transport and trapping, the large extension of the linear range is not a trivial aspect to be implemented.

I cannot but confirm my previous opinion: the proposed device is indeed interesting and novel, and linearization is demonstrated, though on a narrow range. The direct achievement of improved performances with state-of-the-art materials would have raised its impact, removing any doubt regarding the general validity of the concept and the possibility to neatly obtain the working regimes observed for PPV:PCBM with more recent and possibly future organic semiconductor blends. When a semiconductor with improved performances (the DPP copolymer) is used, the linear range is even narrower. This should not be clearly taken as a general rule, but exemplifies my concern.

Overall, I would suggest to publish this work, with all the rest as is, if the authors can show a rationalized improvement of the device performances with a more efficient semiconducting system.

Reviewer #3 (Remarks to the Author):

In this work, a new device architecture, dual gate photo-transistor, is introduced, with the potential to provide a large linear dynamic range, combined with a large photoconductive gain. The concept is certainly original and interesting. The paper has been reviewed by other reviewers, and the authors have addressed most of the concerns. However, I feel that they have only partly succeeded in clarifying the working mechanism. As the paper aims to introduce a new device concept and explain the working mechanism so that it can be improved, it is very important that the exact working mechanism is clear to the reader. I therefore also feel that a figure clarifying the working mechanism belongs in the main text, and not in the supplementary information, as is in this version of the paper. There are still issues unclear in the explanation of the working mechanism. Which carriers are trapped, electrons or holes? Which carriers should reside in long lived traps? Wouldn't trapping reduce the mobility and transit time? For a high gain, the lifetime of the charge carrier contributing to the current should be as long as possible. This lifetime can indeed be made long and constant, by trapping the oppositely charged carrier away from the channel. It's unclear from the text and additional figure R1 if this is the mechanism the authors propose.

To fulfill the premise posed in the introduction, i.e. provide photodetection with a large gain and large

dynamic range so that no external amplification circuits are needed, large improvements are certainly still needed and it is not proven by the authors that this device architecture can fulfill this. Now, an EQE of 170% is rather modest and clearly not sufficient to serve as photodetector without amplification. The linear range is also a modest two decades.

For this reason, I also feel that the paper is very suitable for a specialized journal. The new device architecture has certainly potential, and I am looking forward to the practical implementation of the improvements the authors suggest.

Reviewer #2 comments:

Comment #2-1. The authors answered to my concerns providing very reasonable explanations, and their amendments help to clarify some aspects of their interesting work. Apart from the detail on the bias of the transistor characterized in Fig. S4 (by changing the sign to the drain voltage they are actually inverting the current flow, largely complicating the interpretation of the data), I do not have further technical points.

We thank the reviewer for the positive comment on our previous revision. We have corrected the caption of Fig. S4.

Comment #2-2. The working mechanism, relying on a substantial trapping of one of the two carriers, leaves me some doubt whether the device should be really indicated as ambipolar, or instead as a substantially unipolar device provided with a trapping phase for the complementary carrier.

The reviewer rightly pointed out that while the device operates in presence of both n and p channel (“ambipolar”), the working mechanism still relies on trapping of one of the two carriers. We therefore agree with the reviewer that labelling the device as “ambipolar” may be confusing for the reader, and have revised the manuscript accordingly.

Comment #2-3. The answers confirm that a proof-of-concept dual-gate phototransistor device is proposed for linearization of the photoresponse with intensity, leaving large room for improvement. The PPV:PCBM choice is based on its use by Anthopoulos long ago. It is reasonable, but the achievement of improved performances (such as an extended linearization) is only postulated and is left to further work. Given the complex combination of charge transport and trapping, the large extension of the linear range is not a trivial aspect to be implemented.

We agree with the reviewer that the range of linearization is an important parameter. Our additional results with the PPV:PCBM device show that the linear dynamic range can be improved up to nearly 4 orders by reducing the film thickness such that the channel is further depleted by the top gate (Fig. S7). Although this in turn reduces the gain, the result nevertheless show that it is possible to achieve a wide linear dynamic range using this design. This approach is reasonable given that the gain can be significantly improved by using high mobility organic materials (such as DPP-DTT:PCBM, see response below and Fig. S9).

Fig. S7. MDMO-PPV:PCBM dual-gate phototransistor with improved linear dynamic range. (a) Transfer characteristics measured in dark, showing larger threshold voltage shift by opposite top gate bias (V_{TG}) compared to 30nm device, indicating greater depletion of the bottom channel by the top gate. (b) Increase in source-drain current upon illumination at various intensities. (c) Intensity dependence of photo-responsivity showing increased linearization range compared to 30nm device.

Comment #2-4. I cannot but confirm my previous opinion: the proposed device is indeed interesting and novel, and linearization is demonstrated, though on a narrow range. The direct achievement of improved performances with state-of-the-art materials would have raised its impact, removing any doubt regarding the general validity of the concept and the possibility to neatly obtain the working regimes observed for PPV:PCBM with more recent and possibly future organic semiconductor blends. When a semiconductor with improved performances (the DPP copolymer) is used, the linear range is even narrower. This should not be clearly taken as a general rule, but exemplifies my concern.

We agree with the reviewer that obtaining the working regimes in a high performance material system will help us establish the general validity of the concept. Our additional work on the DPP copolymer based device show that dual-gate operation with improved performance can indeed be achieved by using a high performance organic system, showing linearization at a similar intensity range as the PPV:PCBM device but with much improved responsivity (over 40A/W or $\sim 9000\%$ EQE) due to higher carrier mobility (Fig. S9). Furthermore, this result is consistent with the proposed working mechanism, showing that linearized amplification can be achieved even if the roles of electrons and holes are reversed (DPP system is p-type so electrons are trapped to provide photoconductive gain).

Fig. S9. Improved photo-responsivity of dual-gate phototransistor based on high mobility DPP-DTT:PCBM blend (3:1 weight ratio, operating at V_{BG} of -40V, V_{TG} of +40V and V_D of -5V). The device was illuminated with green light (532nm). The DPP-DTT:PCBM device show similar intensity dependence to MDMO-PPV:PCBM device, showing linearized photoresponse below $\sim 1 \text{ mW cm}^{-2}$. The improved responsivity can be attributed to the higher carrier mobility in the DPP-DTT:PCBM system. Even higher responsivity is expected if the device was illuminated nearer to the absorption peak ($\sim 800\text{nm}$) and with a fully transparent gate to allow maximum light penetration.

Comment #2-5. Overall, I would suggest to publish this work, with all the rest as is, if the authors can show a rationalized improvement of the device performances with a more efficient semiconducting system.

We thank the reviewer for this positive comment. By demonstrating significant improvement in device performances with high mobility organic semiconducting system (DPP copolymer), we believe that this work is now ready for publication.

Reviewer #3 comments:

Comment #3-1. In this work, a new device architecture, dual gate photo-transistor, is introduced, with the potential to provide a large linear dynamic range, combined with a large photoconductive gain. The concept is certainly original and interesting. The paper has been reviewed by other reviewers, and the authors have addressed most of the concerns.

We thank the reviewer for the positive comment.

Comment #3-2. However, I feel that they have only partly succeeded in clarifying the working mechanism. As the paper aims to introduce a new device concept and explain the working mechanism so that it can be improved, it is very important that the exact working mechanism is clear to the reader. I therefore also feel that a figure clarifying the working mechanism belongs in the main text, and not in the supplementary information, as is in this version of the paper.

We agree with the reviewer that it is crucial to explain the working mechanism, and we thank the reviewer for the insightful questions that we need to clarify in the manuscript. For this reason we have provided a detailed explanation of the mechanism in the main text (Lines 151-169 on page 8), with the supporting figure in the SI. Indeed it would be ideal to clarify only in the main text, but due to the complexity of the working mechanism involving carrier transport and trapping, we opt to provide a concise description in the main text and further describe the details in the SI with the help of Figure S8.

Comment #3-3. There are still issues unclear in the explanation of the working mechanism. Which carriers are trapped, electrons or holes? Which carriers should reside in long lived traps? Wouldn't trapping reduce the mobility and transit time? For a high gain, the lifetime of the charge carrier contributing to the current should be as long as possible. This lifetime can indeed be made long and constant, by trapping the oppositely charged carrier away from the channel. It's unclear from the text and additional figure R1 if this is the mechanism the authors propose.

We thank the reviewer for the insightful comment. As described in the revised manuscript, we note that either electrons or holes can be trapped, depending on the system and operation regime. In the case of the PPV:PCBM device, it is n-type so electrons are transported with amplification due to hole trapping. At sufficiently high intensity the deep, long-lived hole traps are filled and additional holes can be transported across the device through the top channel, which results in linear photoresponse with constant gain because the trapping time of holes is not varied by additional light absorption. On the other hand, DPP-DTT:PCBM is p-type so holes are transported with amplification due to electron trapping, and linearization is achieved in the same way as for the PPV device, as we have demonstrated with the additional experiment (Fig. S9). Therefore, the dual-gate device can operate even if the role of the carriers are changed, provided that the two gates have opposite polarity. We have revised the manuscript accordingly in order to clarify this point (Lines 171-183 on page 9).

Comment #3-4. To fulfill the premise posed in the introduction, i.e. provide photodetection with a large gain and large dynamic range so that no external amplification circuits are needed, large improvements are certainly still needed and it is not proven by the authors that this device architecture can fulfill this. Now, an EQE of 170% is rather modest and clearly not sufficient to serve as photodetector without amplification. The linear range is also a modest two decades. For this reason, I also feel that the paper is very suitable for a specialized journal. The new device architecture has certainly potential, and I am looking forward to the practical implementation of the improvements the authors suggest.

We agree with the reviewer that our prototype device based on PPV:PCBM shows rather modest performance in terms of both gain and linear dynamic range, and further improvement would further support the practicality of this design. We believe that our additional results help us confirm that the device performance can indeed be improved, both in terms of gain (up to ~9000% EQE as shown in the DPP device; Fig S9) and linear dynamic range (up to 4 decades by further channel depletion; Fig S7). Thus, our work provides the required framework for realizing high-performance dual-gate phototransistors in future studies.

REVIEWERS' COMMENTS:

Reviewer #2 (Remarks to the Author):

The authors further revised their work and underlined how their proof-of-concept can be improved by adopting higher mobility materials. This answers my concern on the applicability of the new detecting concept. I am of the opinion that now the manuscript should be published in Nature Communications.

I'd like only to point out a couple of very minor points to be corrected before publication:

- On page 11, "Beta" is not defined, nor it is present in the noise expression.
- Panel e in Fig. 1 and Panels a,b,c in Fig. S8 have graphical problems.

Reviewer #3 (Remarks to the Author):

The authors have addressed all my and the other referees comments adequately and have significantly improved the manuscript. The addition of data on a higher performing material system certainly strengthens arguments on the potential of this new concept. I recommend publication.

Reviewer #2 comments:

Comment #2-1. The authors further revised their work and underlined how their proof-of-concept can be improved by adopting higher mobility materials. This answers my concern on the applicability of the new detecting concept. I am of the opinion that now the manuscript should be published in Nature Communications.

We thank the reviewer for the positive comment on our previous revision.

Comment #2-2. I'd like only to point out a couple of very minor points to be corrected before publication:

- On page 11, "Beta" is not defined, nor it is present in the noise expression.

We have corrected this by replacing "beta" to "slope".

- Panel e in Fig. 1 and Panels a,b,c in Fig. S8 have graphical problems.

We have fixed the graphic issues in Fig. 1 and Fig. S8.

Reviewer #3 comments:

Comment #3-1. The authors have addressed all my and the other referees comments adequately and have significantly improved the manuscript. The addition of data on a higher performing material system certainly strengthens arguments on the potential of this new concept. I recommend publication.

We thank the reviewer for the positive comment.